# Evaluation of Machine Learning Algorithms for Surface Water Extraction in a Landsat 8 Scene of Nepal [note 1]

**DOI:** 10.3390/s19122769

**Published:** 2019-06-20

**Authors:** Tri Dev Acharya, Anoj Subedi, Dong Ha Lee

**Affiliations:** 1Institute of Industrial Technology, Kangwon National University, Chuncheon 24341, Korea; tridevacharya@kangwon.ac.kr; 2Department of Civil Engineering, Kangwon National University, Chuncheon 24341, Korea; 3School of Geomatics and Urban Spatial Information, Beijing University of Civil Engineering and Architecture, Beijing 102616, China; 4Institute of Forestry, Pokhara Campus, Tribhuvan University, Pokhara 33700, Nepal; anojsubedi99@gmail.com

**Keywords:** surface water mapping, machine learning, naive Bayes, recursive partitioning and regression trees, neural networks, support vector machines, random forest, gradient boosted machines, Landsat, Nepal

## Abstract

With over 6000 rivers and 5358 lakes, surface water is one of the most important resources in Nepal. However, the quantity and quality of Nepal’s rivers and lakes are decreasing due to human activities and climate change. Despite the advancement of remote sensing technology and the availability of open access data and tools, the monitoring and surface water extraction works has not been carried out in Nepal. Single or multiple water index methods have been applied in the extraction of surface water with satisfactory results. Extending our previous study, the authors evaluated six different machine learning algorithms: Naive Bayes (NB), recursive partitioning and regression trees (RPART), neural networks (NNET), support vector machines (SVM), random forest (RF), and gradient boosted machines (GBM) to extract surface water in Nepal. With three secondary bands, slope, NDVI and NDWI, the algorithms were evaluated for performance with the addition of extra information. As a result, all the applied machine learning algorithms, except NB and RPART, showed good performance. RF showed overall accuracy (OA) and kappa coefficient (Kappa) of 1 for the all the multiband data with the reference dataset, followed by GBM, NNET, and SVM in metrics. The performances were better in the hilly regions and flat lands, but not well in the Himalayas with ice, snow and shadows, and the addition of slope and NDWI showed improvement in the results. Adding single secondary bands is better than adding multiple in most algorithms except NNET. From current and previous studies, it is recommended to separate any study area with and without snow or low and high elevation, then apply machine learning algorithms in original Landsat data or with the addition of slopes or NDWI for better performance.

## 1. Introduction

Nepal is a geographically diverse country with flats in the south and increasing hills, to the mighty Himalayas in the north. In Nepal, approximately 70% to 90% of the total annual rainfall occurs during the monsoon period resulting in high runoff and sediment discharge causing surface water area change [1]. Thus, it is rich in water resources with approximately 600 rivers [2] and 5358 lakes [3]. Due to such seasonal variation and large surface water area, it is difficult to track changes in surface water [4,5]. Furthermore, the change in stream-flows due to climate change has also been predicted [6,7]. Therefore, the monitoring and estimation of surface water is an essential task.

In such cases, remote sensing technology plays a very important role on detecting, extracting, and monitoring surface water [8,9]. Open and free access mid-resolution multi-spectral satellite images such as Landsat brings further benefits in the process [10]. Thus, the authors begin to utilize the Landsat database to extract surface water from a small case of Phewa to a Landsat scene that covers different types of surface water along with features that resemble water, such as shadows, forests, built-ups, snow and clouds. In previous studies, the authors evaluated water index methods, single and combined [11,12], along with the segmentation of the scene [13]. Our latest work showed promising results for a scene in which segmentation and the optimum threshold were manually identified based on the given set of reference dataset. As a next step, automated extraction of surface water with well-known supervised classification approaches were evaluated [14].

With recent developments in computing technology, the machines are cheaper, and the algorithms are efficient. Therefore, the abundance of these machines and machine learning algorithms have been widely applicable in almost every aspect of human life. Moreover, their optimization has outperformed the classical ones. Numerous machine learning algorithms have been applied for remotely sensed imageries [15,16,17,18,19]. These algorithms can be divided broadly into three categories: (a) Unsupervised learning; (b) supervised learning; and (c) reinforcement learning. Unsupervised learning groups are given an unlabeled dataset based on the implicit relationship/function. Supervised learning utilizes a certain labeled instance (training dataset) to predict a similar dataset. Reinforcement learning does not provide a precise label, rather it takes the next step based on the goal-oriented feedback available for each prediction. Reinforcement learning is still in the developing stage, and as there are no errors, it could be wrong with each positive reward. Results of unsupervised learning cannot be ascertained and can be less accurate, whereas supervised provides specific class and labels with better accuracy [20]. Some of the most common supervised algorithms are decision trees, naive Bayes (NB), neural networks (NNET), regression, support vector machines (SVM), and ensemble methods [21]. The libraries for these algorithms have been well developed and implemented in reliable ecosystems of open source tools, such as Python and R languages. Despite the availability of open access data and tools, the evaluation of such in Nepal has never been documented. Moreover, the challenge of varying conditions in a single scene is also new for testing the performance of machine learning algorithms in the extraction of surface water.

Hence, the motivation of this work is to introduce the application of the most common algorithms used by the remote sensing community in Nepal and evaluate their performance for surface water extraction. The six most common algorithms, naive Bayes (NB), recursive partitioning and regression trees (RPART), neural networks (NNET), support vector machine (SVM), random forest (RF), and gradient boosted machines (GBM) were evaluated in a Landsat 8 operational land imager (OLI) bands. Also, the slope, normalized difference vegetation index (NDWI) and normalized difference water index (NDWI) were combined one at a time and all three at once with OLI bands to evaluate whether the combination can overcome the limitations of the original bands in water extraction. In the future, such evaluation will assist in selection of proper methods to develop an effective time series database at national scale in Nepal.

## 2. Materials and Methods

As the study is the extension of our previous study [13], the authors utilized the same Landsat scene and reference dataset in this study. Hence, details on the study area and data can be found in Acharya et al. [13]. Pre-processing of the Landsat scene was carried out in Environment for Visualizing Images (ENVI) version 5.3 (Exelis Visual Information Solutions, Boulder, CO, USA), cartographic maps were produced in ArcGIS 10.5 (Environmental Systems Research Institute, California, CA, USA), and the machine learning process were carried using Classification And REgression Training (CARET) package in R 3.5.0 (The R Foundation, Vienna, Austria) software packages.

### 2.1. Cases

To better evaluate and compare the results, our previous study had six classes in the scene [13]. However, it was found that Himalayan areas with shadows and melting ice are quite challenging in the task. Hence, the authors introduced two more cases that will help better understand the results. Figure 1 shows the Landsat 8 scene in pansharpened natural color composite image with water and non-water reference dataset and cases.

### 2.2. Methods

Using the digital elevation model (DEM) and pre-possessed OLI bands (LS8), three secondary bands were created: Slope, NDVI and NDWI. Figure 2 shows the secondary bands. After which these three secondary bands were stacked one by one with LS8 such that this study has three new datasets as LS8 + Slope, LS8 + NDVI, and LS + NDWI. Finally, all three were stacked with LS8 to form another new data LS8 + Slope + NDVI + NDWI.

A total of 800 reference datasets were used in the whole scene. To minimize the overfitting or selection bias in the predictive ability of machine learning algorithms, the authors used ten folds cross-validation to train the model. In the process, the data is divided into ten sets, nine sets are first used to train the model and the remaining one is used for validation purposes. The process is repeated ten times with each of the ten subsamples being used for both training and validation. Thus, each observation is used exactly once for training and validation.

After forming the dataset, six machine learning methods, NB, RPART, NNET, SVM, RF, and GBM, were implemented in the CARET package of R, and were used for training and preparing the models. The advantage of the CARET package is that it provides complete machine learning processes from data preparation, training, modelling and prediction to validation. Moreover, the train function in the caret sets up a grid of tuning parameters as required for the algorithm which fits each model and calculates a resampling based performance measure. As details on these models can be widely found in the literature, the authors will be briefly describing these algorithms.

NB is a probabilistic classifier based on Bayes’ theorem with the independent assumptions between predictors. It is easy to build, with no complicated iterative parameter estimation and suited particularly when the dimensionality of the inputs is high. With few tuneable parameters and fast, they end up being very useful as a quick-and-dirty baseline for a classification problem. 

RPART is a type of binary tree used for classification or regression tasks. It performs a search over all possible splits by maximizing an information measure of node impurity, selecting the covariate showing the best split.

NNET in R, is a feed-forward neural network with a single hidden layer flowing left to right. Feed-forward neural networks were the first type of artificial neural network invented, and are simpler than their counterparts, recurrent neural networks. They are called feed-forward networks because information only travels forward in the network (no loops), first through the input nodes, then through the hidden nodes (if present), and finally through the output nodes. These are primarily used for supervised learning in cases where the data to be learned are neither sequential nor time dependent.

SVM is a data classification method that separates data using a hyperplane. In other words, for the given labelled training data (supervised learning), the algorithm outputs an optimal hyperplane which separates only one type of data. The SVM technique is generally useful for data that is non-regularity, which means data whose distribution is unknown.

RF is a meta estimator that fits several decision tree classifiers on various sub-samples of the dataset and uses averaging to improve the predictive accuracy and control over-fitting.

GBM is a class of ensemble learning techniques to create a collection of shallow and weak successive trees with each tree learning and improving on the previous based on a cost function (for example, squared error).

All these six models for five different datasets were applied to the full scene to classify the image into binary water and non-water maps. After the classification, the full reference dataset was evaluated for overall accuracy (OA) and kappa coefficient (Kappa) [22].

## 3. Results and Discussion

In this section, first, the results of the cross-validated models were assessed, and then they were applied to the whole scene for the surface water extraction. Next, the complete reference dataset was tested against the predicted results. In the end, a detailed comparison and discussion of the different machine learning algorithms were done to evaluate the performance case by case.

Figure 3 shows the boxplot of the OA and Kappa for different models based on the cross-validated data for five different multiband data. At first glance, it can be seen that, other than RPART and ND, four algorithms showed quite high OA and Kappa in the cross-validation. In the case with only LS8 multiband data, NNET and RF were able to achieve maximum OA and Kappa. After adding the slope, NNET and RF produced the model with maximum OA and Kappa. However, adding NDVI and NDWI with the LS8 bands, only NNET achieved improved OA and Kappa. Moreover, it can be clearly seen that the OAs achieved, with all the machine learning algorithms, narrow ranges with the addition of NDWI, followed by NDVI and the slope. The mean of OA and Kappa shifted towards higher values after adding only the slope and all three secondary bands for all algorithms, except RPART.

In the CARET package, the train function produces the model with tuning parameters. Therefore, the produced models can be easier to apply in the prediction. Figure 4 shows the predicted result of all the algorithm models for all the multiband data. Based on the visual inspection, all the algorithms performed well in the lower flat and hilly regions compared to the Himalayas. Adding the slope made the result visually better in most algorithms, except the RPART. Similarly, adding only NDWI and adding all three secondary bands produced visually better results. In contrast, NDVI only improved with RF and GBM algorithms and others produced many misclassified surface water bodies. For further evaluation on how well all these methods have performed in the scene, eight different types of cases were carefully analyzed and compared for each algorithm in the section below.

For the quantitative evaluation, the confusion matrix-based OA and Kappa were produced for the full reference class and predicted class. In Figure 5, both OA and Kappa shows similar patterns for all the algorithms in response to the addition of the secondary band. RF shows both OA and Kappa 1 for all multiband data. Following this, GBM, NNET and SVM performed well. However, the NB and RPART performance were among the worst. In NB, LS8 and LS + NDWI data did well compared to others. The addition of NDVI or the slope or all three secondary bands decreased the performance. While in the case of RPART, the addition of one or all three bands improved the OA and Kappa. In NNET, the addition of all bands increased the performance compared to each secondary band, while it was opposite for the SVM i.e., a decrease in performance. In GBM, adding NDWI boosted the performance the highest, and the slope and all three did well, but the addition of NDVI decreased the performance.

Except RF and NNET, similar patterns of performance were seen. NDWI followed by the slope, increases the performance of the machine learning algorithms. In contrast, NDVI or adding all bands reduced the performance. It shows that adding specific secondary bands with the original LS8 band is useful for enhancement of surface water. Similarly, the slope that ensures flat surface of water bodies is also useful if added. In NDVI, water are negative values at the same time that many other non-vegetative bodes can also have the same value, which has led to a decrease in performance. NNET and SVM are very well-known state-of-art algorithms in classification. They perform well in many cases and their performance in the original Landsat 8 scene of Nepal is satisfactory. However, the interesting result is that while NNET increased the performance with that addition of data gradually, SVM went the other side. This also shows that the addition of secondary data does not necessarily improve performance and are rather dependent on algorithms. Thus, comparative studies are necessary to check these performances and select the best one for the area under study.

Figure 6 shows the cases for NB in the entire multiband, along with the slope and results from the previous study, i.e., conditioning NDVI and NDWI with elevation (Elev_NDWnVI) [13]. Cases a, g and h were the ones representing snow and melting ice in that Himalayas, in which NB performed very bad and most of the shadowed regions were misclassified. It performed well in cases from b to f, and even did well than in the previous study in case c, i.e., narrow river channels with shadows. It was able to classify non-water shadow features well. In the case of additional band performance, all the results in the cases are similar with no distinct difference. As per the RPART performance in cases (Figure 7), it was able to remove the shadow issues in the narrow river with shadows in case b. It performed well in cases of wide shadows, ponds and wide rivers. However, it was not able to separate shadows of the Himalayas in cases a, g and h. Further, adding the slope somewhat improved these cases but failed in the case of the shadow in case b. However, adding NDVI and NDWI were unable to improve in the shadows.

In Figure 8, NNET showed gradual improvement in identifying water bodies with the addition of secondary bands. NDWI enhanced the shadows as water compared to NDWI and all three together. For case b, the addition of NDVI misclassified few cloud shadows while others did not. For case c, all the dataset were able to solve the shadow issue from the previous shadow in the hills, except the NDVI. For the plain area in case d with the narrow river, NNET results were good in all the dataset with even enhanced small water bodies. In case e for urban small water bodies, LS8 predicted water and some shadows but with the addition of secondary bands, the pure pixels were enhanced. With the addition of all three bands, it refined the water bodies very well. For a large river in case f, the results are similar to each other and with the previous study [13]. For cases g and h, the results are different for every dataset. The slope misclassified the melting ice, NDVI misclassified the shadows and combined, showed both misclassified. LS8 only and the addition of NDWI were somewhat better compared to others. With the SVM in Figure 9, the SVM showed similar results to NNET for most cases. However, SVM misclassified the darkest shadows as water in the whole scene. It is clearly seen in case a, b, g and h, especially with the addition of the slope. In case a, with LS8 and addition of NDVI and NDWI, even light shadows were misclassified.

As both RF and GBM are ensemble methods, their performance on the cases are quite similar in Figure 10 and Figure 11 respectively. Both methods were successful in separating shadows in case c, but failed in cases a and g. In addition, with the addition of the slope only, the results were better than the original LS8. However, the addition of NDVI or NDWI did not perform well in the shadow areas. Only the addition of the slope as a secondary band seems to be a good choice rather than adding all for both RF and GBM algorithms in surface water mapping.

In comparison to the previous study [13], except NB and RPART, all other results are above 90% and even up to 100% as seen in the literature [16,23,24,25]. In addition, for a hilly test scene in the study done by Jiang et al., [15] using a multilayer perceptron neural network in Landsat 8 OLI satellite imagery, the neural network performance was similar to this work, i.e., OA 98.50 and Kappa 0.970. Compared to our previous study [13], the results were good for hilly and lower flatlands where there is no snow. The performance of surface water detection for narrow rivers in hilly regions with shadows improved, however, they were only able to detect pure water pixels. The main issues were the ice and snow with shadows in hilly areas. Adding the slope somewhat improved cases in the Himalayas, but not the NDVI or NDWI. In a case by case evaluation also, the performance seem well but were mostly misclassified in the Himalayas. Quantitatively, machine learning algorithms were much better compared to the index methods, however, the results are not as reliable as it should be for the whole scene. In the full scene, very few areas are covered by the snow compared to the large hilly and flat lands. Nevertheless, Nepal is a mountainous country and has a quantitatively large cover, which could be a challenge in applying machine learning algorithms. Thus, the wrong predictions in those areas are less significant in the validation. A further investigation with cutting-edge machine learning technology, i.e., convolutional neural networks or deep learning could be undertaken for improvement.

## 4. Conclusions

In this study, we extended the previous study and applied six machine learning algorithms: NB, RPART, NNET, SVM, RF and GBM to evaluate the surface water extraction using a Landsat 8 OLI images in Nepal. Using the previous reference dataset and Landsat scene, six different models were developed using the CARET package in R software. Cross-validation was completed to minimize the overfitting then train the model to predict the surface water and validate the full reference dataset. With three secondary bands: Slope, NDVI and NDWI, the algorithms were evaluated for performance with the addition of extra information. The results were compared, case by case, and the following conclusions were drawn from the test scene and applied machine learning algorithms:(a)All the applied machine learning algorithms showed OA above 90% but in case of Kappa except NB and RPART, it was above 90%.(b)RF showed OA and Kappa both 1 for the all the multiband data with the reference dataset.(c)GBM and NNET also showed good performance followed by the SVM.(d)Machine learning algorithms were able to perform better in the hilly regions and flat lands but not well in the Himalayas with ice, snow and shadows.(e)The addition of the slope and NDWI showed improvement in the results compared to the NDVI except NNET. Others do not improve with the addition of all three secondary bands compared with the individual addition.

It seems that machine learning methods could be very useful for the accurate automated binary classification of surface water in Nepal. The use of RF with original LS8 data or with the addition of the slope or NDWI with another algorithm can be undertaken. Based on this and previous work [13], it is recommended to segment the study area with and without snow or low and high elevation, then apply RF or GBM for better performance.

For further investigation, this study aimed to evaluate the application of convolutional neural networks or deep learning for better accuracy. In addition, individual original bands and secondary bands with the RF and GBM can be evaluated so that high accuracy can be achieved with minimum bands.

## Figures and Tables

**Figure 1 sensors-19-02769-f001:**
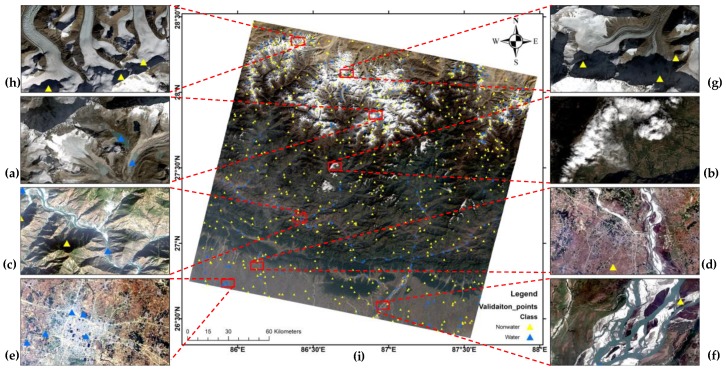
Pansharpened Landsat 8 natural color composite image with water and non-water reference dataset; (**i**) Each red box represents different surface water bodies and noises case in the scene (**a**–**h**).

**Figure 2 sensors-19-02769-f002:**
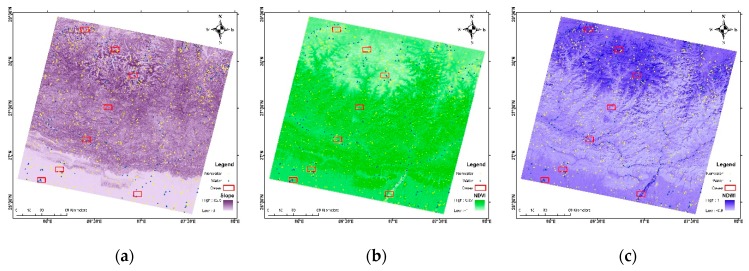
Secondary bands (**a**) slope; (**b**) NDVI and (**c**) NDWI.

**Figure 3 sensors-19-02769-f003:**
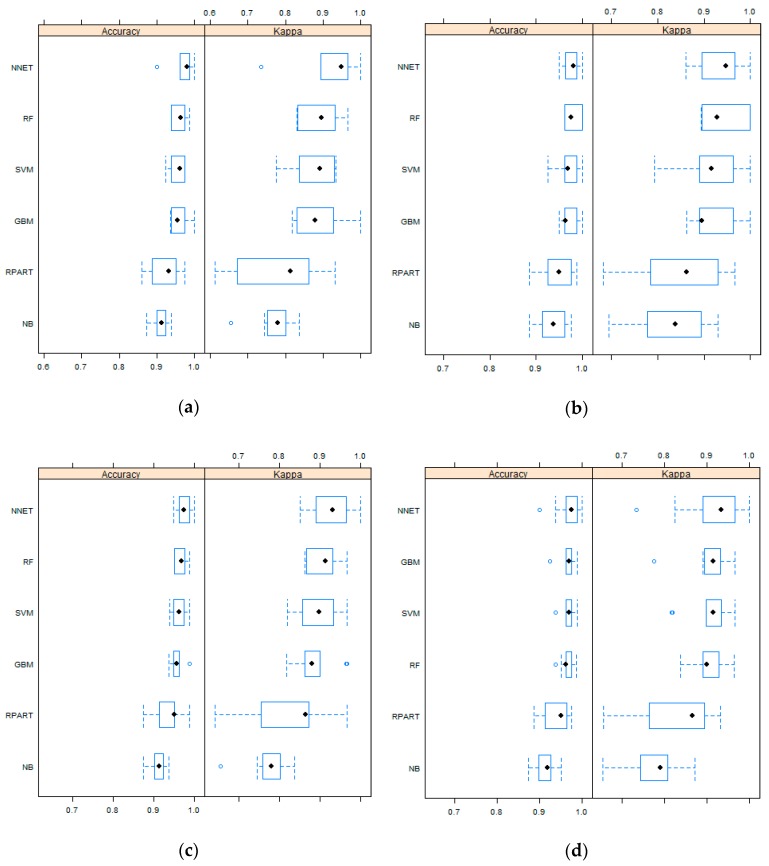
The boxplot of overall accuracy (OA) and the Kappa coefficient (Kappa) using different machine learning algorithms for provided multiband data: (**a**) LS8; (**b**) LS8 + Slope; (**c**) LS8 + NDVI; (**d**) LS8 + NDWI; and (**e**) LS8 + Slope + NDVI + NDWI.

**Figure 4 sensors-19-02769-f004:**
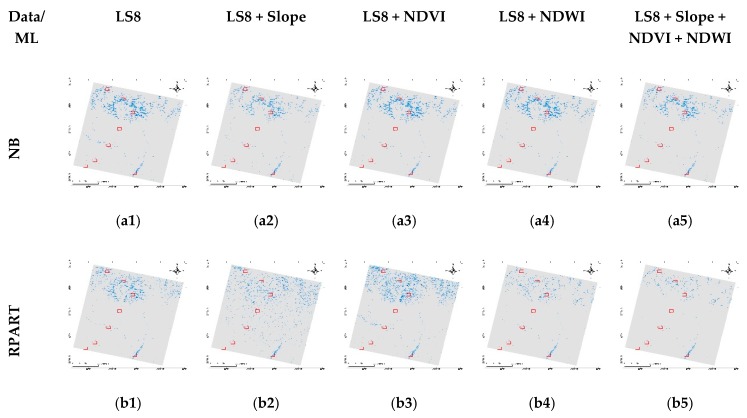
Results of surface water extraction using different machine learning methods (**a**–**f**) for provided multiband data (**1**–**5**).

**Figure 5 sensors-19-02769-f005:**
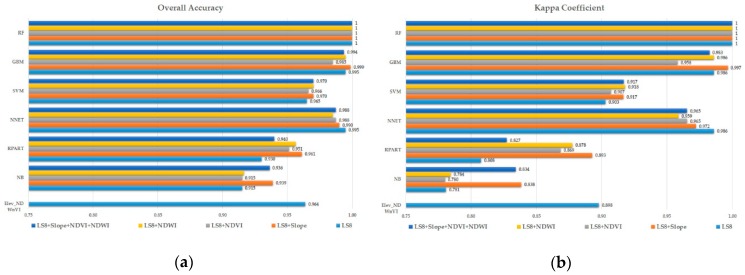
(**a**) Overall accuracy (OA) and (**b**) Kappa coefficient (Kappa) of all the algorithms for provided multiband data.

**Figure 6 sensors-19-02769-f006:**
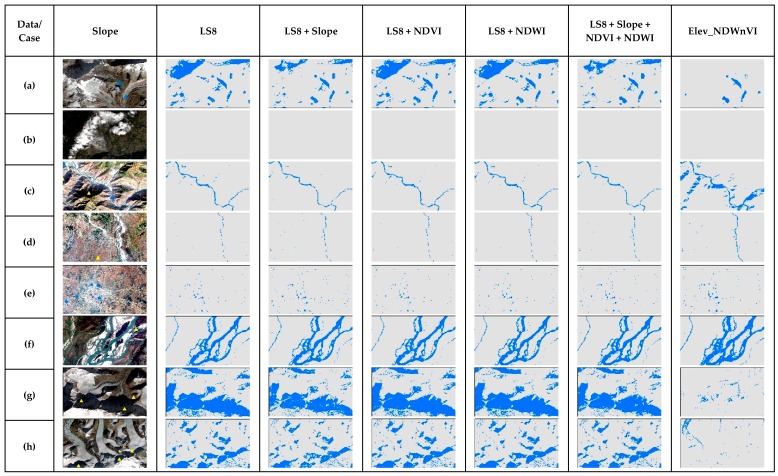
The comparison of NB results of special cases of surface water in the test scene (Figure 1) for different multiband data used.

**Figure 7 sensors-19-02769-f007:**
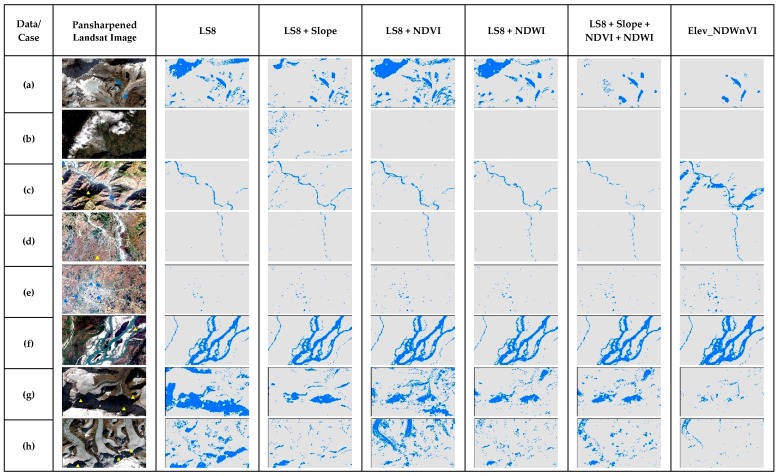
The comparison of RPART results of special cases of surface water in the test scene (Figure 1) for different multiband data used.

**Figure 8 sensors-19-02769-f008:**
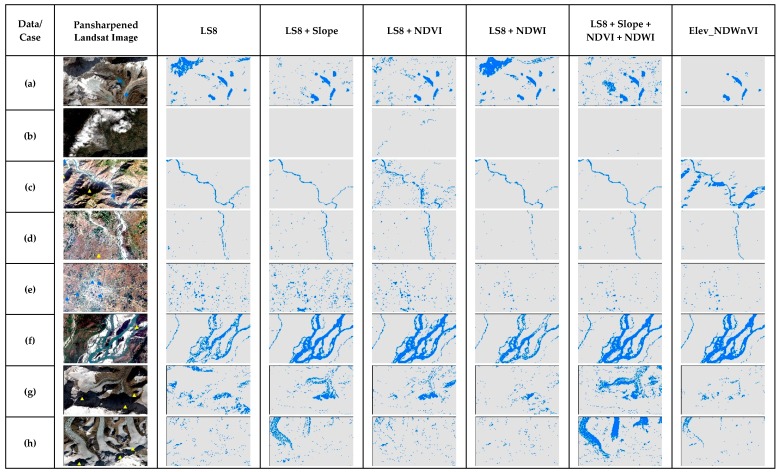
The comparison of NNET results of special cases of surface water in the test scene (Figure 1) for different multiband data used.

**Figure 9 sensors-19-02769-f009:**
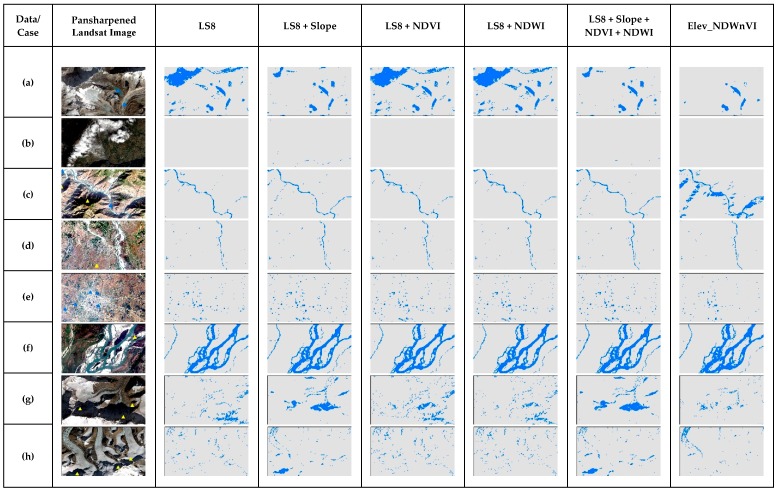
The comparison of SVM results of special cases of surface water in the test scene (Figure 2) for different multiband data used.

**Figure 10 sensors-19-02769-f010:**
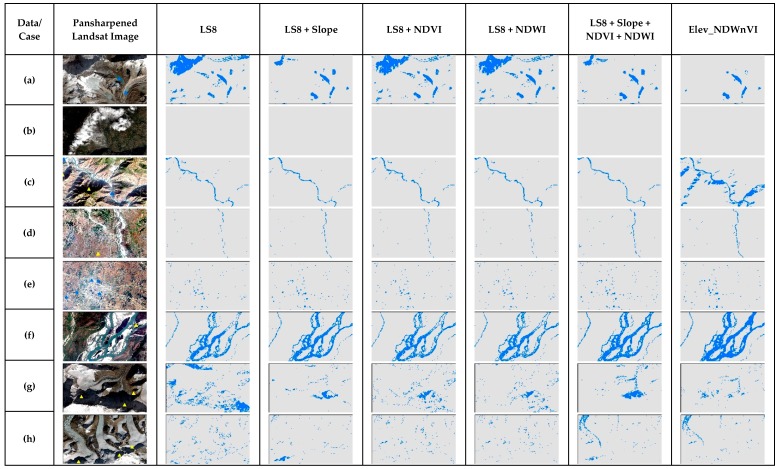
The comparison of RF results of special cases of surface water in the test scene (Figure 1) for different multiband data used.

**Figure 11 sensors-19-02769-f011:**
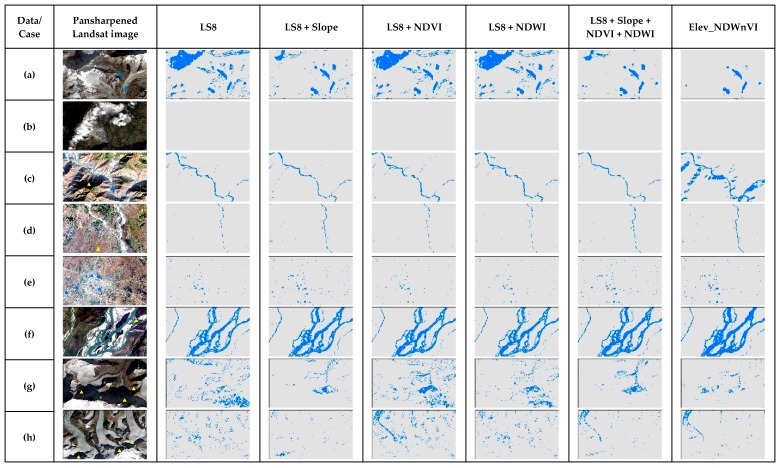
The comparison of GBM results of special cases of surface water in the test scene (Figure 1) for different multiband data used.

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
