# Peer review of "Evaluation of Machine Learning Algorithms for Surface Water Extraction in a Landsat 8 Scene of Nepal"

_sensors, 2019, doi:10.3390/s19122769_

Reviewer 1 Report

This manuscript presents an evaluation research on water extraction with a variety of machine learning algorithms and indices. Experimental results shown in this manuscript could be an interesting focus in the readership of remote sensing community. However, several issues should be addressed to meet the requirement of Sensor journal.

I list my comments as follows,

1. An evaluation on machine learning algorithms must include convolutional neural networks, which is an edge-cutting machine learning technology.

2. Raw satellite images might be more appropriate in Figure 6 and 7, instead of NDVI and slope map.

3. This manuscript includes not only different machine learning algorithms, but also different indices. Thus, the authors may consider a new title.

4. (a)-(e) in the Conclusion Section is not valid for a paper focusing on machine learning algorithms. Extensive explanations are required here. Generally speaking, SVM and RF outperformed other machine learning algorithms (not deep learning algorithms).

However, things could be different under different occasions. For example, some machine learning algorithms could perform better than deep learning algorithms based on limited number of training samples. Moreover, features (e.g. LS8, LS8+slope, etc.) used for learning are also critical for evaluating the performance of different machine learning algorithms. Generally speaking, people prefer to create sparse features to feed a machine learning model. But, this common phenomenon might vary according to the data used for classification task.

Author Response

Point 1: Dear Authors,

This manuscript presents an evaluation research on water extraction with a variety of machine learning algorithms and indices. Experimental results shown in this manuscript could be an interesting focus in the readership of remote sensing community. However, several issues should be addressed to meet the requirement of Sensor journal.

I list my comments as follows,

1. An evaluation on machine learning algorithms must include convolutional neural networks, which is an edge-cutting machine learning technology.

Response 1: Dear reviewer,

First thank you very much for taking out time and reviewing the manuscript. This is our second test work to prepare national database of surface water in Nepal. As the country is diverse geographically, we found problem in implementing single index or threshold. Hence, we started learning and prepared the previous work with water indices (Acharya et al., 2018). However, recently Machine Learning Methods (MLMs) are quite a buzz and much studies are carried out with them for such classification or regression works. Hence, we did this second experiment on the test scene.

Even though previous study was good, special cases were still not well classified, which were well handled by MLMs. As a conclusion from these two works, we plan to segment the whole Nepal and use MLMs to develop national surface water database automated and as frequent as possible to monitor the change.

We totally agree with reviewer that Convolutional Neural Networks (CNNs) or deep learning are the cutting edge MLMs and lags behind Multilayer Perceptron (MLP). And these must be tested for the scene. We were trying the Tensorflow programming but were not successful so far and used simple feed-forward neural network. And also found that (Jiang et al., 2018) used MLP Neural Network for Surface Water Extraction in Landsat 8 OLI Satellite Imagery. Their scene h was similar to ours and accuracy. As it is still not CNNs, we will keep your comment in mind to perform the convolutional neural networks or Deep learning in next work.

As this is extended manuscript for special issue of ECSA5, the Editorial Board only allows one round of revision process in order to achieve rapid turnaround with the deadline for publication within 10 days, we apologize that we are unable to add new analysis in short time frame.

Point 2: 2. Raw satellite images might be more appropriate in Figure 6 and 7, instead of NDVI and slope map.

Response 2: Thank you for the suggestion. We have replaced the first figure with satellite image in Figs 5-7.

Point 3: This manuscript includes not only different machine learning algorithms, but also different indices. Thus, the authors may consider a new title.

Response 3: We think this work is continuation of the past work and the outcome is the combined results of both of these studies. Hence, the title were kept similar with previous one but the discussion included the previous best result for comparison and result especially recommendation at last paragraph is the outcome of the both studies.

We believe that as previous paper was only with indices, this paper based on MLMs result is well represented by the title.

Point 4: (a)-(e) in the Conclusion Section is not valid for a paper focusing on machine learning algorithms. Extensive explanations are required here. Generally speaking, SVM and RF outperformed other machine learning algorithms (not deep learning algorithms).

However, things could be different under different occasions. For example, some machine learning algorithms could perform better than deep learning algorithms based on limited number of training samples. Moreover, features (e.g. LS8, LS8+slope, etc.) used for learning are also critical for evaluating the performance of different machine learning algorithms. Generally speaking, people prefer to create sparse features to feed a machine learning model. But, this common phenomenon might vary according to the data used for classification task.

Response 4: We are extremely sorry, if we mislead the conclusion with missing words. We have added: “The results were compared cases by case and following conclusions were drawn for the test scene and applied machine learning algorithms:” and “Based on this and previous work (Acharya et al., 2018), it is recommended to segment study area” to make them more specific to the study and MLMs.

We totally agree with the reviewer’s view. Everything is relative and most of the studies we also found are quite surprising good. In recent days, advancing computer technology lead to much studies with 90-99% accuracies without knowing the actual field/phenomena. This work is was designed when we found global databases that report high accuracy are not matching the ground reality in Nepal. Thus, we selected this unique scene and did special cases to understand the underlying problems.

There are still areas where we were unable to reach and verify, the sample points are still lacking the sparse characteristics of the waters. Recently we realized rather than classifying, probability of prediction is better way to understand the surface water and non-water areas in imagery and based on such further segmented classification can improve the classification for all types of water for whole Nepal or earth. Although simple, the work is our personal effort and could be very helpful for Nepal and its researchers.

Hope we are successful in convincing the reviewer the motivation and importance of this work. One again, thank you for your time and considerations.

References

Acharya, T.D., Subedi, A. and Lee, D.H. (2018). "Evaluation of Water Indices for Surface Water Extraction in a Landsat 8 Scene of Nepal." Sensors, Vol. 8, No. 7, pp. 2580.

Jiang, W., He, G., Long, T., Ni, Y., Liu, H., Peng, Y., Lv, K. and Wang, G. (2018). "Multilayer perceptron neural network for surface water extraction in Landsat 8 OLI satellite images." Remote Sensing, Vol. 10, No. 5, pp. 755.

Reviewer 2 Report

This is an extension of a previous research by the authors. The paper presented different machine learning methods for the water surface extraction from a Landsat 8 image.

I have some general comments regarding the study here.

1.       In the previous study, the authors claimed an OA of 0.96 with only the index based methods. The machine learning here then has only little room for improvements. This paper has also got similar conclusions to the previous one that we need to segment the elevation and the snow present/absence to achieve a better accuracy, then what is the major contribution of this paper?

2.       The authors have only applied all the methods on one landsat 8 scene, have you tried to apply the methods on other locations for a better verification?

3.       The setup of the machine learning methods is too brief, could the authors show more about how many samples used for the ten fold cross-validation, how many samples are used for testing for the accuracy evaluation? What is the computing performance for each method take to classify the image?

Author Response

Point 1:

This is an extension of a previous research by the authors. The paper presented different machine learning methods for the water surface extraction from a Landsat 8 image.

I have some general comments regarding the study here.

1. In the previous study, the authors claimed an OA of 0.96 with only the index based methods. The machine learning here then has only little room for improvements. This paper has also got similar conclusions to the previous one that we need to segment the elevation and the snow present/absence to achieve a better accuracy, then what is the major contribution of this paper?

Response 1: First thank you very much for taking out time and reviewing. We really appreciate that you understood the manuscript and linked it with the previous study of ours.

This is our second test work to prepare national database of surface water in Nepal. As the country is diverse geographically, we found problem in implementing single index or threshold. Hence, we started learning and prepared this work. We learned that combining indices increases the separability and segmenting (changed as per other reviewer suggestion) with elevation can make it more precise (Acharya et al., 2018). However, recently Machine Learning Methods (MLMs) are quite a buzz and much studies are carried out with them for such classification or regression works. Hence, we did this second experiment on the test scene. Even though previous study was good with 0.96, special cases were still not well classified, which were well handled by MLMs. As a conclusion from these two works, we plan to segment the whole Nepal bassoon elevation and use MLMs to develop national surface water database automated and as frequent as possible to monitor the change.

We are extremely sorry, if we mislead the conclusion with missing words. We have added: “The results were compared cases by case and following conclusions were drawn for the test scene and applied machine learning algorithms:” and “Based on this and previous work (Acharya et al., 2018), it is recommended to segment study area” to make them more specific to the study and MLMs.

We think this work is continuation of the past work and the outcome is the combined results of both of these studies. Hence, the title were kept similar with previous one but the discussion included the previous best result for comparison and result especially recommendation at last paragraph is the outcome of the both studies.

Point 2: The authors have only applied all the methods on one landsat 8 scene, have you tried to apply the methods on other locations for a better verification?

Response 2: As stated earlier, our main aim is to make national database of Nepal. We are doing it based on our personal capacity. Hence, we choose the best test scene that represents whole Nepal. For better representation of this work, we used ten folds cross-validation method.

As an extended manuscript of special issue of ECSA5, there was deadline. Thus, we believed it can be published for now with the test scene and lay foundation for national scale work.

Few researchers form Nepal got interested with our previous work and are volunteering with us to collect national wide verification points, we hope to utilize the results from both this and previous study to implement whole Nepal with multiple scenes in near future.

Point 3: The setup of the machine learning methods is too brief, could the authors show more about how many samples used for the ten fold cross-validation, how many samples are used for testing for the accuracy evaluation? What is the computing performance for each method take to classify the image?

Response 3: Sorry for the confusion. But from our previous reviews, most of the reviewers pointed not to explain too much and too deep mathematically in application based works unless it is novel or rarely used methods. In our case, the methods are universally known MLMs and there are plenty of detailed papers. We explained in simple words pointing their understanding only.

As the study is the extension of our previous study, we utilized same Landsat scene and reference dataset in this study. Hence, details on study area and data can be found in Acharya et. al. (2018).  However, to make it clear, we added a line stating, “A total of 800 reference dataset were used in the whole scene.” in the beginning of the paragraph to make it clear.

As the name states and it is very commonly known about ten-fold cross-validation in which,  9/10th dataset i.e. 720 points will be used for training and remaining 1/10th i.e. 80 will be for validation until all 10 sets are finished. We thought it wold be more obvious. As it is only a scene based case study, we believe computation cost in terms of data, speed and accuracy are not very important. However, we will keep that in mind for national scale work.

In Nepal, geospatial research is still nascent phase. There are now undergraduate courses but are mostly producing surveyors. Lack of government priority thus lack of research directions. While developed world has EOS and GEE cloud computing done in few seconds, these results does not match with ground reality in countries like Nepal. We faced this challenged and started this work based on personal effort. This case study could not only improve mapping methodologies but also motivate young researchers in Nepal.

Hope we are successful in convincing the reviewer the motivation and importance of this work. One again, thank you for your time and considerations.

References

Acharya, T.D., Subedi, A. and Lee, D.H. (2018). "Evaluation of Water Indices for Surface Water Extraction in a Landsat 8 Scene of Nepal." Sensors, Vol. 8, No. 7, pp. 2580

Round  2

Reviewer 2 Report

I would like to thank the authors for addressing my comments. This paper has some overlap to the authors' previous study, but I think there is still some contribution from the methodology perspective and this paper is an intermediate step to the national scale work. I look forward to future work and interested in how the proposed methods work with a large scale of data.